# Comparative MD simulations and advanced analytics based studies on wild-type and hot-spot mutant A59G HRas

**Neeru Sharma, Uddhavesh Sonavane\*, Rajendra Joshi** *

HPC–Medical and Bioinformatics Applications Group, Centre for Development of Advanced Computing, Pune, India

* uddhaveshs@cdac.in (US); rajendra@cdac.in (RJ)

**Data Availability Statement:** All relevant data are within the manuscript and its Supporting Information files.

## Abstract

The Ras family of proteins is known to play an important role in cellular signal transduction. The oncoprotein Ras is also found to be mutated in ~90% of the pancreatic cancers, of which G12V, G13V, A59G and Q61L are the known hot-spot mutants. These ubiquitous proteins fall in the family of G-proteins, and hence switches between active GTP bound and inactive GDP bound states, which is hindered in most of its oncogenic mutant counterparts. Moreover, Ras being a GTPase has an intrinsic property to hydrolyze GTP to GDP, which is obstructed due to mutations and lends the mutants stuck in constitutively active state leading to oncogenic behavior. In this regard, the present study aims to understand the dynamics involved in the hot-spot mutant A59G-Ras using long 10µs classical MD simulations (5µs for each of the wild-type and mutant systems) and comparing the same with its wild-type counterpart. Advanced analytics using Markov State Model (MSM) based approach has been deployed to comparatively understand the transition path for the wild-type and mutant systems. Roles of crucial residues like Tyr32, Gln61 and Tyr64 have also been established using multivariate PCA analyses. Furthermore, this multivariate PCA analysis also provides crucial features which may be used as reaction coordinates for biased simulations for further studies. The absence of formation of pre-hydrolysis network is also reported for the mutant conformation, using the distance-based analyses (between crucial residues) of the conserved regions. The implications of this study strengthen the hypothesis that the disruption of the pre-hydrolysis network in the mutant A59G ensemble might lead to permanently active oncogenic conformation in the mutant conformers.

## Introduction

The RAS (RAt Sarcoma) is a protein encoded by a crucial proto-oncogene, which is found to be mutated in ~25% of human cancers. The Ras protein has three isoforms namely: N-Ras, H-Ras and K-Ras where the catalytic domains have 90% sequence identity and the major difference is confined only in the so-called hypervariable region (HVR) at the C-terminal [1]. The distinct HVR of the C-terminal region is responsible for different interactions with the

**Funding:** The Authors received funding from National Supercomputing Mission (NSM), Govt. of India

**Competing interests:** The authors have declared that no competing interests exist.

**Abbreviations:** RAS, RAt Sarcoma; HRas, Harvey Rat sarcoma; GTP, Guanosine Tri-Phosphate; MD, Molecular Dynamics; SwI, Switch-1; SwII, Switch-2; P-loop, Phosphate-binding loop; GBR, GEF-Binding Region.

membrane and differential accessibility to the activator proteins as well. The differences in the C-terminal region amongst these isoforms lead to distinct post translational modifications which ultimately lead to characteristic biological functions of the different isoforms [2]. In normal conditions, Ras is known to be associated with a range of cellular processes like cell-proliferation, cell-differentiation, apoptosis and plays a regulatory role in cytoplasmic signaling networks [3]. It is a well established fact that Ras functions as a nodal point in signal transduction to control cell-proliferation, migration and neuronal activity [4–6]. The Ras protein also plays a pivotal role in downstream signalling of the associated effector molecules and membrane localization [7]. The best characterized and most studied Ras effectors are Raf proteins, which belong to the family of Serine/Threonine kinases [8]. The Ras protein is known to adopt distinct conformational states, including multiple transient states, where majority of the local changes are notably present in GEF-binding region (GBR), switch 1 (SwI) and switch 2 (SwII) [6, 9–12]. Moreover being a GTPase, the Ras family of proteins possess an intrinsic property to hydrolyze the associated GTP (Guanosine Tri-Phosphate) molecule to GDP (Guanosine Di-Phosphate) molecule [13]. The Ras proteins basically are G-proteins, which inherently function as a binary signalling switch with "ON" and "OFF" states [14, 15]. The GTP bound active conformation of the Ras protein is referred as the "ON" state while the GDP bound inactive conformation is referred as the "OFF"state. Moreover, Ras isoforms have a considerably low energetic barrier between these active "ON" and inactive "OFF" states [16]. The continuously programmed switching between the active and inactive states is assisted by two classes of proteins namely: GEF (Guanine-nucleotide Exchange Factor) and GAP (GTPase Activating Protein) [17, 18]. On one hand, GEF specifically helps in Ras activation by replacing GDP with the GTP, while GAP catalyzes the process of GTP hydrolysis by hydrolyzing the associated GTP molecules [19, 20]. The role of GAP in catalyzing the GTP hydrolysis process by forming a "two-water model" network is also well established [21–23].

The loss of intrinsic GTP hydrolysis activity or becoming insensitive towards external GAP proteins is a prominent feature of the oncogenic gain-of-function mutations [24]. Any single amino acid mutation at the sequence level may lead to subtle structural variation but pronounced functional differences in the mutant structures as compared to the wild-type structures [25]. Gain-of-function Ras mutations are found to be present in almost 25% of human cancers and due to different biological functions served by different Ras isoforms RAS-isoform/effector and RAS-mutation-specific therapeutic approach must be explored to achieve improved therapeutic effects [3]. Efforts are, hence, going on to design isoform and mutant specific Ras inhibitors which could efficiently increase the therapeutic window [26–28]. Also, the mutational variants of the Ras protein are prone to get stuck in the active state as the respective activation energies are increased in mutant conformations [29, 30]. Hence, exploring the dynamics of the mutants and comparing the same with the wild-type is required to address the mutational effects on the protein structure, as a single residue substitution might also have dramatic effects on the protein dynamics. Hence, many studies have focussed on targeting this RAS pathway, understanding the underlying mutants for cancer therapy and designing better therapeutics to treat such disorders [31, 32]. The studies have aimed at understanding the mechanism of Ras signalling and how they are related to human disorders including cancer. Furthermore, correlating the structural, functional and associated dynamics is quintessential for related future investigation including drug repurposing as well drug development based studies [1, 24]. After the availability of the crystal structure of Ras protein, initial simulation studies were focussed only on the individual nucleotide states of the wild-type and mutant conformations [33, 34]. Later on, specific studies on the conserved RAS regions were reported where conformational states of Sw I region were studied in detail, where residues Val29 and Ile36 were specifically mutated to glycines [35]. Research encompassing classical

and advanced MD simulation studies on various hot-spot Ras mutants like G12V, G13V, A59G, Q61L, Q61H have gained acceleration since the last decade.

Earlier, many classical MD, QM/MM simulation studies were carried out on wild-type and mutant Ras systems to understand the associated dynamics. Such studies explore the structural and dynamical aspect of the crucial regions of the Ras protein including switch regions (SwI and SwII) and few loop regions as well [22, 36]. The studies could also explain the distinct conformational states of conserved regions SwI and SwII in GDP and GTP bound states [37, 38]. A study demonstrating the milliseconds time-scale activation phenomenon of the wild-type Ras activation was also reported recently [39]. Role of conserved crystal waters was also reported in this study. Similarly, another study demonstrating the role of conserved water molecules in Q61H K-Ras mutants has also been reported [40]. A restrained and free MD simulation spanning1.76 μs simulation length has been reported in the presence and absence of selected crystallographic water molecules. The study also demonstrated that the presence and absence of these waters made the protein to sample distinct conformations and such water molecules act as allosteric ligands to induce a population shift. A similar MD simulation study has reported the dynamic behavior of A59G H-Ras mutants as compared to the wild-type H-Ras [25]. The study also reported increased flexibility in the conserved switch regions along with loop 3, helix 3 and loop 7 regions and a lower energetic barrier between GTP and GDP bound conformations in the A59G H-Ras. Another crucial residue of the Ras protein is Q61, which plays an important role in GAP assisted GTP hydrolysis. Also, the role of Gln61 along with Glu63 and water molecules has also been reported for the GTP hydrolysis process in Ras conformers [41]. Specific hydrogen bond interactions of residues Thr35, Gly60 and Lys16 have also been reported.

Mutations in the Ras proteins are most frequently found at G12, G13 and Q61 positions. In this regard, a comprehensive 6.4 μs of MD simulation study demonstrated the mechanism underlying the GAP mediated GTP hydrolysis process for the mutants at G12, G13 and Q61 positions [1]. In the reported study, G12V, G12C, G12D, G13D and Q61H mutants of K-Ras were investigated for their respective mechanisms and differential oncogenicity. Differently impaired GAP mediated hydrolysis activity was reported in this study for the mutants and previously distinct intrinsic GTP hydrolysis active was also reported in NMR-based functional profiling based report for Ras mutations [42]. This suggests an overall impairment of the GTP hydrolysis (intrinsic as well as GAP mediated) in the hot-spot missense Ras mutants. Recently, a QM/MM and QM/MM-MD simulation study on the G12V and G13V Ras mutations was also reported [43]. Gln61 residue and conserved crystal waters were found to be displaced in G12V mutants while G13V mutants showed enhanced flexibility in the pre-hydrolysis network formed by Gln61, GTP,water molecules and Arg789-GAP. Both the mutants ultimately encountered a slower hydrolysis reaction and suggest that residue Gln61 is directly involved in the chemical transformations involved in GTP hydrolysis. The pivotal role of Gln61 and its interaction with GAP protein is also established, though QM/MM based *in silico* studies [44, 45]. Role of crucial residues (Thr35, Asp57, Gly60 and Gln61) of the conserved regions from SwI and SwII, especially in maintaining the pre-hydrolysis state has been reported in previous studies [39, 46].

Another hot-spot gain-of-function mutation in the conserved switch region (SwII) is A59G, which plays a crucial role in local conformational changes of the Ras protein [47]. A 20-ns unbiased MD simulation demonstrated a GTP-to-GDP conformational transition by removing γ phosphate of the bound GTP from A59G HRas [25]. The study also depicted that A59G HRas is intrinsically more dynamic than wild-type HRas and mutant HRas conformation has a lower energy barrier between GTP and GDP bound states, as compared to its wild-type counterpart. More recently, a 15 μs MD simulation study on the wild-type and mutant

A59G HRas was recently reported [48]. The study emphasized on the role of Gln61 residue and the absence of pre-hydrolysis state in mutant A59G HRas systems. The study suggested a decreased rate of intrinsic hydrolysis as well as GAP-mediated hydrolysis by making the movement of SwII region highly restricted. In this regard, the current study presents a detailed insight into the dynamics involved in the mutant A59G HRas conformations. Advanced analysis techniques like MSM based analytics, multivariate PCA analysis and MM-GBSA energy calculations were deployed to draw comparative conclusions for the wild-type and mutant simulation trajectories. In addition, interactions between relevant crucial residues from the P-loop (GBR), SwI and SwII regions were also calculated.

The crystal structure of wild-type HRas with PDB id 1QRA [21] and for mutant A59G HRas with PDB id 1LF0 [47] were available in RCSB Protein Data Bank [49]. The two structures were taken for simulation studies with GNP replaced with GTP in mutant structure and a cumulative ~10 μs MD simulations (5μs each for wild-type and mutant) were carried out. The analysis of interactions of the crucial residues of the pre-hydrolysis network of the wild-type Ras suggested important role of Tyr32, Thr35, Gln61 and Tyr64 residues, hence these features were chosen to perform feature based PCA analysis, along with the energy component.

## Materials and methods

### System preparation for MD simulations

The crystal structures of the wild-type (GTP-bound) and mutant A59G-HRas (GNP-bound) were downloaded from RCSB-PDB database with PDB ids: 1QRA and 1LF0, respectively. To maintain the uniformity in the respective systems and the simulation protocols, GNP from the mutant A59G-H-Ras structure was replaced with the GTP molecule. The start structure for these classical MD simulation trajectories were captured from the wild-type and mutant A59G ensembles from the well-tempered metadynamics simulation from the previous study [48].

### MD simulations

All-atom classical MD simulations were carried out for these GTP-bound systems using GRO-MACS [50] with AMBERFF99SB force-field [51]. A time step of 2 fs was used for production MD runs with short-range van der Waals cut-off kept at 14 Å. The water model chosen was Simple Point Charge (SPC) and coordinates were saved at every 2 ps. FFT optimization and PME algorithms were used to calculate electrostatic terms. An energy minimization of 50,000 steps was performed initially and further, NVT (volume) and NPT (pressure) equilibrations were carried out for 1ns each. Modified Berendsen thermostat for maintaining the simulation temperature and Parrinello-Rahman coupling for the pressure were used. The production runs for wild-type WT-H-Ras-GTP-Mg$^{2+}$ and mutant A59G-H-Ras-GTP-Mg$^{2+}$ complexes were then performed. As GTP being a non-standard molecule and its parameters not available in gromacs, its parameters were calculated from the Prodrg server [52].For the present study, ~10 μs of classical MD simulations were performed for the wild-type and mutant A59G H-Ras systems namely *simulationW* and *simulationM*, respectively. The MD simulations were performed using in-house PARAM-BRAF supercomputing cluster at C-DAC.

### Visualization and analysis

For structure visualization, molecular graphics and analyses, tools like UCSF Chimera [53] and VMD (Visual Molecular Dynamics) [54] were used. Xmgrace [55] was used to plot the two-dimensional plots of the relevant analysis carried out. Detailed analyses were carried out for these simulations, namely interactions of the residues belonging to conserved switch

regions, multivariate PCA (using selected features) using RStudio package, MMGBSA based energy calculations, Markov State Model (MSM) based analysis of molecular kinetics and thermodynamic models from MD data using PyEMMA [56].

## MSM based analyses

With respect to MSM, the creation of states and the transition matrix depends on the choice of collective variable (CVs) or features and the lag time [57–60]. In order to build the MSM model for the MD simulation data in this study, following steps were carried out: (a) VAMP Score Calculation: This was done for different features, which is based on the variational principle [61]. The variational principle helps to check the robustness of MSM. VAMP scores were calculated for following features viz. dihedral angles, residue minimum distance between 2 groups of residues (mindist) and the distances between groups with respect to their C-alpha atoms (Cadist). (b) Feature Selection (CV): The Cadist feature was then used as the feature for further MSM calculations. (c) Time Independent Component Analysis (TICA): Using the selected feature i.e. Cadist, TICA calculation was performed in the second step. TICA helps to reduce the dimensionality of the trajectory data. (d) Clustering: The MD simulation data for all the trajectories was then clustered using K-means clustering algorithm for both the wild-type and mutant systems. (e) Macrostates Creation: In the next step, Principal Components & Classification Analysis (PCCA) calculation was performed and macrostates were created for the respective MD simulation clusters (f) Transition Matrix: Finally, the transition matrix was then built using the chosen lag time in such a way that the transition matrix follows markovian behaviour. The eigen decomposition of the transition matrix results in eigenvectors which represents slow motions of the systems. The fluxes and mean passage time calculations were performed by computing the transition pathways between the states using the discrete transition path theory [62]. The choice of the lag time is related to eigenvalue which gives us a physically measurable timescale known as implied timescale ($t_i$) using Eq 1:

$$t_i = -\tau/(\ln\lambda_i) \tag{Eq1}$$

where $t_i$ is implied timescale, $\tau$ is the lag time and $\lambda_i$ is an eigenvalue.

The validation of the MSM was done using Chapman Kolmogorov equation [57] given by the Eq 2:

$$T(n\tau) = T(\tau)^n \tag{Eq2}$$

where n is an integer representing the number of steps, $T(\tau)$ is a transition matrix at lag time $\tau$.

The above relationships help in validating the chosen MSM model, which ultimately plays a crucial role in determining the transient intermediate states as well the involved kinetics amongst them. All these steps of MSM model preparation and analysis were carried out for each of the systems under study.

## Results

### Basic analyses: Root Mean Square Deviation (RMSD) and Root Mean Square Fluctuation (RMSF)

**RMSD.** Basic RMSD analysis was carried out for *simulationW* and *simulationM* of the wild-type and mutant simulation trajectories, respectively. The RMSD calculation was done for the protein backbone and respective conserved regions (P-loop, SwI and SwII) with respect to the respective first frame. RMSD plots for wild-type Ras showed more deviation in case of wild-type trajectories as compared to the mutant A59G system, as shown in Fig 1. Fig 1

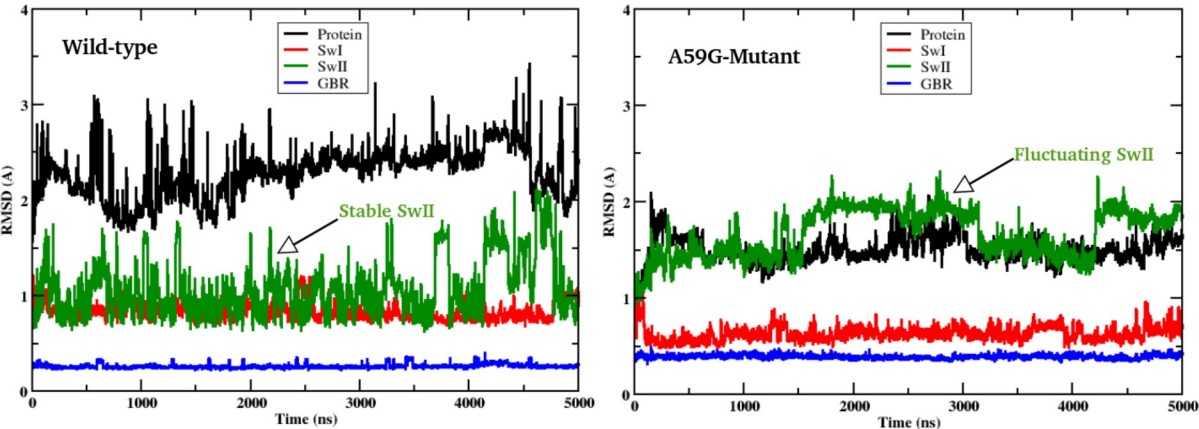

**Fig 1. 2D-plot showing RMSD trend for the wild-type and mutant *simulationW* and *simulationM*, respectively.** The mutant system showed a fluctuating SwII region as compared to its wild-type counterpart.

showed a deviation for protein backbone of 2.5–3 Å for wild-type and 1.5–2 Å for the mutant, respectively. The GBR and SwI region did not show major deviation, with respective RMSD values of 0.25 Å and 0.5 Å for GBR and 0.75 Å and 0.5 Å for SwI (for wild-type and mutant systems, respectively). Fig 1 also showed an increased deviation in case of mutant simulation for SwII region (especially the loop region, L4) where the RMSD value laid between 1.5–1.75 Å, as compared to 0.75 to 1 Å for the wild-type simulation. The RMSD suggests a notable change in SwII region as compared to the wild-type counterpart, which is also reflected in the complete protein backbone's RMSD values.

**RMSF.** RMSF was also calculated for the two systems. Fig 2 shows the RMSF plot *simulationW* and *simulationM*. For the residues flanking the SwI region (residues 25–29 and residues

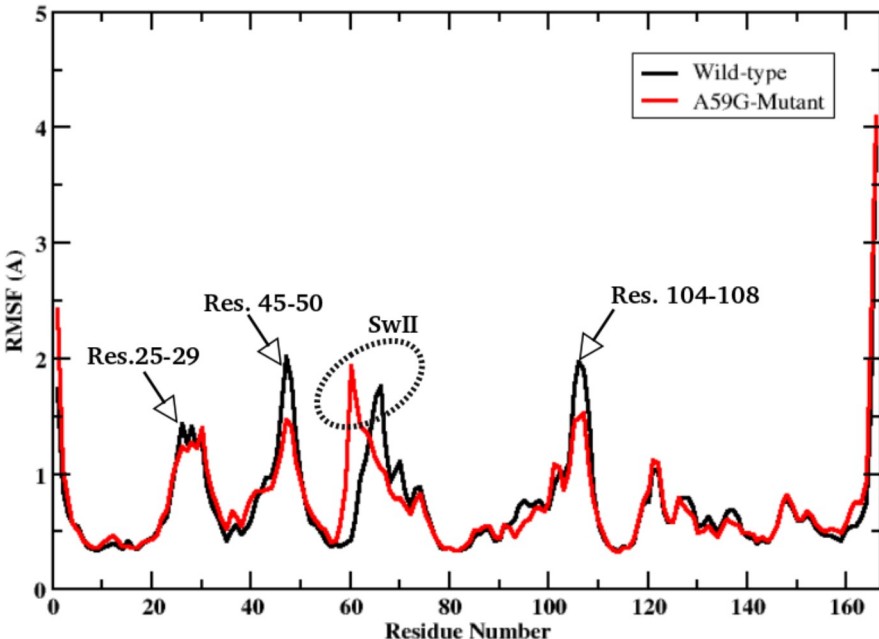

**Fig 2. 2D-plot showing RMSF trend for the wild-type and mutant *simulationW* (black line) and *simulationM* (red line), respectively.** The regions showing changes are labelled accordingly.

45–50), a difference of 0.5 Å was observed for the wild-type and mutant trajectories. Also, the RMSF trend of the SwII region, especially the loop region (residues 59 to 65) showed distinct trends for the wild-type and mutant simulations. Interestingly, the region between residues 45–50 and 104–108 also showed an decreased RMSF variation (of ~0.5 Å) in mutant simulation as compared to its wild-type counterpart. As these deviations are small in quantity, structural analysis of the respective trajectories was also performed to understand these changes in more detail. As depicted in Fig 3, the highlighted labels show the localised variations for above three mentioned regions. The relative thicknesses of the respective regions show the distinct variation throughout the 5 μs of the respective simulation trajectories. The above mentioned residue stretches (25–29, 45–50, 59–65 and 104–108) showed clear distinction for the wild-type and mutant systems, as depicted by the corresponding ribbon thickness.

### Analyses for conserved GBR, SwI and SwII regions

**Interactions of crucial residues of the conserved regions.** The basic RMSD and RMSF analyses suggested distinct differences in GBR, SwI and SwII regions. Hence, crucial residues from these regions were then checked for their interactions with other residues as follows: Lys16, Ser17 (GBR), Tyr32, Thr35 (SwI), Asp57, Gly60, Gln61 and Tyr64 (SwII). Figs 4–6 show the plots for these interactions as the number of conformers (for 5 μs of the respective trajectories) versus distance between the respective pairs. Residue Gln61 showed distinct interactions with Lys16, Ser17, Thr35 and Tyr64 for the two trajectories. For this, the simulation length is represented as the number of conformers and the respective distance bin for each residue pair.

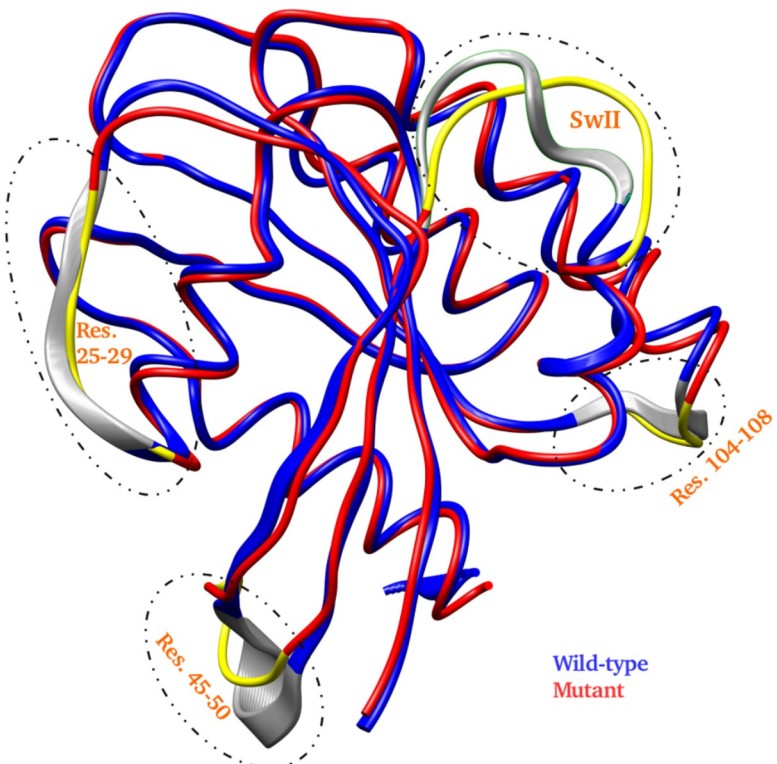

**Fig 3. Overlaps of the wild-type and mutant simulation trajectories to depict the deviation in the selected regions in dotted ovals (gray for wild-type and yellow for A59G mutant).** The wild-type and mutant trajectories are shown in blue and red cartoon representations, respectively.

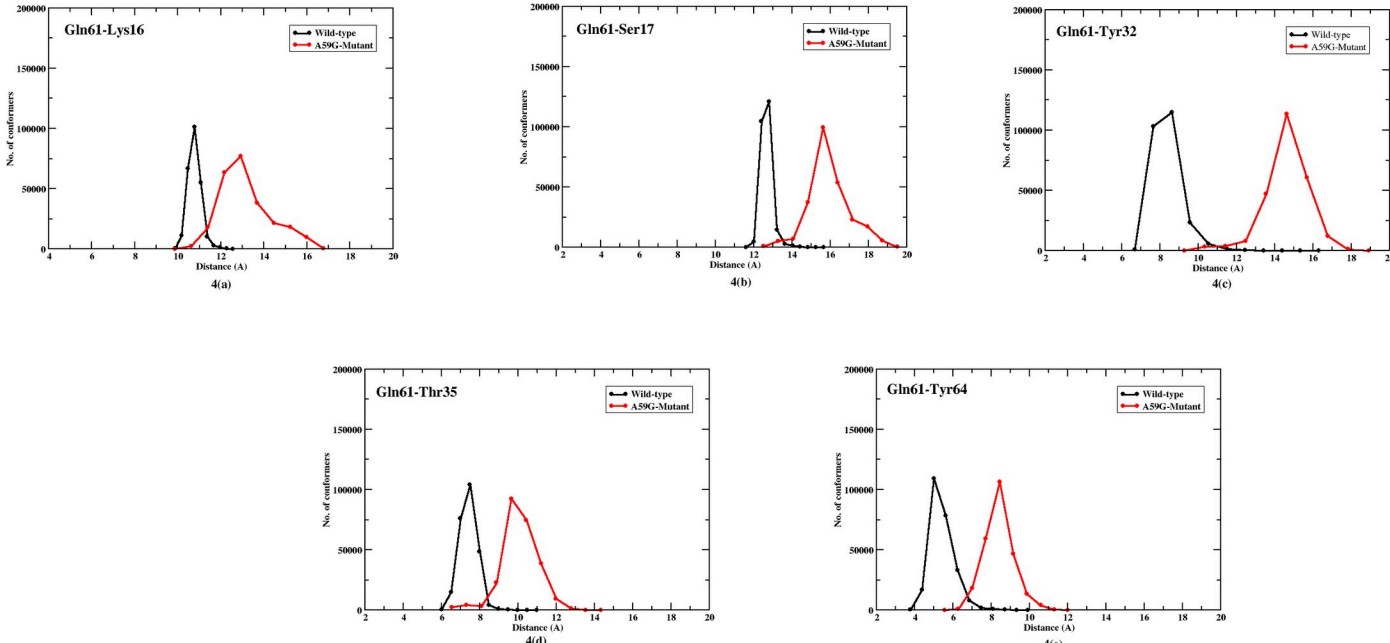

**Fig 4. 2-D plot showing number of conformers for Gln61 residue's interaction with Lys16, Ser17, Thr35, Tyr32 and Tyr64.** Black and red lines represent trends for the wild-type and mutant systems, respectively.

Firstly in the wild-type *simulationW*, for the Gln61-Lys16 pair, the interaction laid in range 10–12 Å with maximum conformation concentrated at 11 Å and mutant *simulationM* had this interaction in range of 12–17 Å with an average 13 Å, as shown in Fig 4(A). Similarly for the Gln61-Ser17 pair, Fig 4(B) showed that the *simulationW* showed maximum conformations in 12–13.5 Å and the mutant *simulationM* had a spread out conformations in 14–19 Å range. The third interaction for the wild-type simulation showed a distance of 6–8 Å and mutant simulation at 8–13 Å for the Gln61-Thr35 residue pair, as depicted in Fig 4(C). The residues Tyr32 and Tyr64 showed distinct interaction trends with Gln61. Fig 4(D) demonstrated that the pair namely: Gln61-Tyr32, showed significant difference in the distance range for the wild-type and mutant systems. For the wild-type simulation, the interaction fell in the 7–11 Å range and for mutant simulation the residues were more distant at a distance of 12–18 Å. The most crucial difference of Gln61 interaction was found with Tyr64 (as shown in Fig 4(E)), where for the wild-type system the residues were facing each other (as visualized in the structural

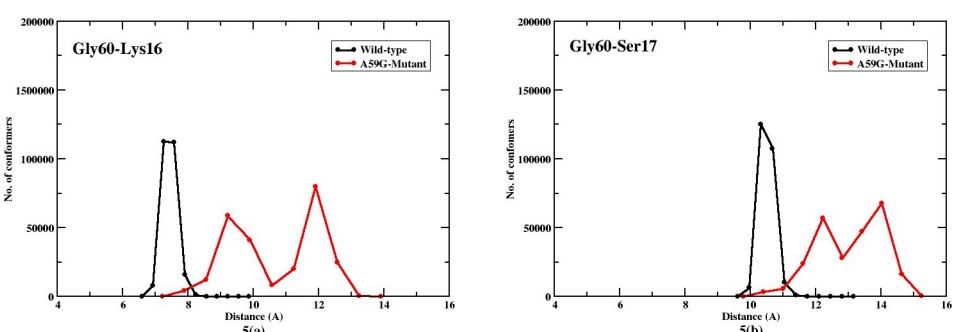

**Fig 5. 2-D plot showing number of conformers for Gly60 residue's interaction with Lys16, and Ser17.** Black and red lines represent trends for the wild-type and mutant systems, respectively.

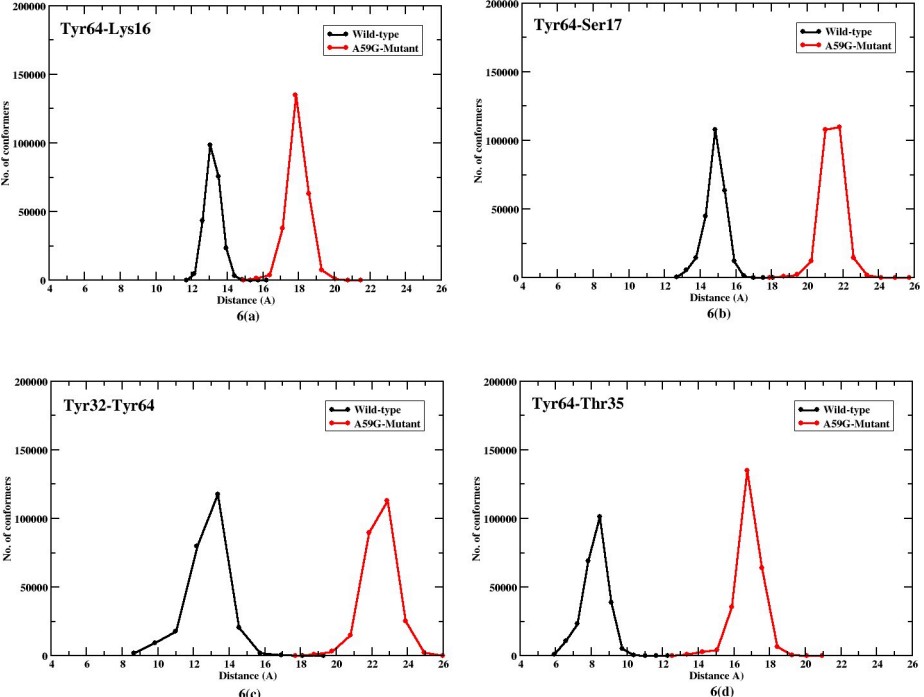

**Fig 6. 2-D plot showing number of conformers for Tyr64 residue's interaction with Lys16, Ser17, Thr35 and Tyr32.** Black and red lines represent trends for the wild-type and mutant systems, respectively.

analysis) within range 4–7 Å, while for the mutant system the residues were an increased distance of 6–11 Å, due to a change in Gln61 orientation.

Further, residue Gly60 was also analyzed for its interaction with other residues of the conserved regions. As residue Gly60 directly plays a crucial role in GTP hydrolysis for Ras systems, its interaction was analyzed for the residues belonging to Lys16 and Ser17 (the conserved residues of GBR). Majorly Lys16 and Ser17 showed distinct interaction in wild-type and mutant simulations. Fig 5(A) and 5(B) showed plots for Gly60-Lys16 and Gly60-Ser17 respectively. For the wild-type simulation, Gly60-Lys16 mostly fall in 6.5–8 Å and 8–13 Å for the mutant simulation. A similar trend was observed for Gly60-Ser17 interaction as well, where the wild-type simulation had interaction in range 10–11 Å and between 11–15 Å for the mutant counterpart. The residue Gly60 showed fluctuating trends with its interaction with GTP, due to absence of conserved water molecules needed for maintaining pre-hydrolysis state (plots not shown). Interestingly, two maxima were observed for these interactions in the mutant A59G simulations owing to the reason that Gly60 showed increased fluctuations in mutant *simulationM*.

Similarly, residue Tyr64 was also checked for its interaction with Lys16, Ser17, Tyr32 and Thr35, of which Lys16 and Ser17 (GBR) showed a similar trend while Tyr32 and Thr35 (SwI) share a typical trend of their respective interactions with Tyr64. Fig 6(A) and 6(B) show Tyr64-Lys16 and Tyr64-Ser17 interactions. The wild-type simulation showed a range of 12–14 Å and 13–17 Å and mutant simulations with 16–20 Å and 12–19 Å, for Tyr64-Lys16 and Tyr64-Ser17 interactions respectively. Interestingly, interactions with Tyr32 and Thr35 segregated the wild-type and mutant conformation in two separated clusters. Fig 6(C) depicts that the Tyr64-Tyr32 interaction in wild-type was in the range 9–15 Å and 20–25 Å for the mutant system. Similarly for the wild-type system, the Tyr64-Thr35 interaction was in range 6–10 Å

and for mutant it was 12–19 Å, as shown in Fig 6(D). The comparative trends of these crucial interactions suggest a significant effect on the orientation and distinct behavior of the SwII residues, especially Gly60, Gln61 and Tyr64.

**Dihedral analyses of Tyr64 residue.** As interaction of Tyr64 with residues of SwI region (Tyr32 and Thr35) showed distinguishable characteristics for wild-type and mutant trajectories, a visual inspection of the Tyr64 side chain was carried out for both the trajectories. The orientation of Tyr64 (along with neighboring Gln61 residue) differed in the wild-type and mutant trajectories. To quantify this change, dihedral angle analysis was carried out for Tyr64 residue as shown in Fig 7. For Φ dihedral, both the systems laid in -30˚ to -150˚. On the other hand, the ψ dihedral showed clear distinction between wild-type and mutant simulations. For the wild-type system, the ψ angle laid in -60˚ to 60˚ while for the mutant counterpart, the ψ dihedral had a different orientation between 60˚ to 120˚.

On the basis of above analyses, it is established that the residues Gln61 and Tyr64 majorly differentiates between the wild-type and A59G mutant ensembles.

**Multivariate / Feature-based principal component analysis.** Collating these residues and the associated features like distances, dihedral and energy component (S1 File), a multivariate PCA analysis was performed. The preliminary analyses helped in searching the available search space for all the potential features and then finalizing the most informative ones for the multivariate PCA. The major rationale behind performing PCA is to determine low-dimensional variates, in this case amongst many interactions, angles, dihedrals etc like features/factors. For this, scaling of all the input features was performed first which might have different variances for distinct individual features. After this standardization of the variables involved, the selected feature-based PCA was performed using the R statistical package. The features (variables) for this PCA analysis were chosen on the basis of previous analyses for the two

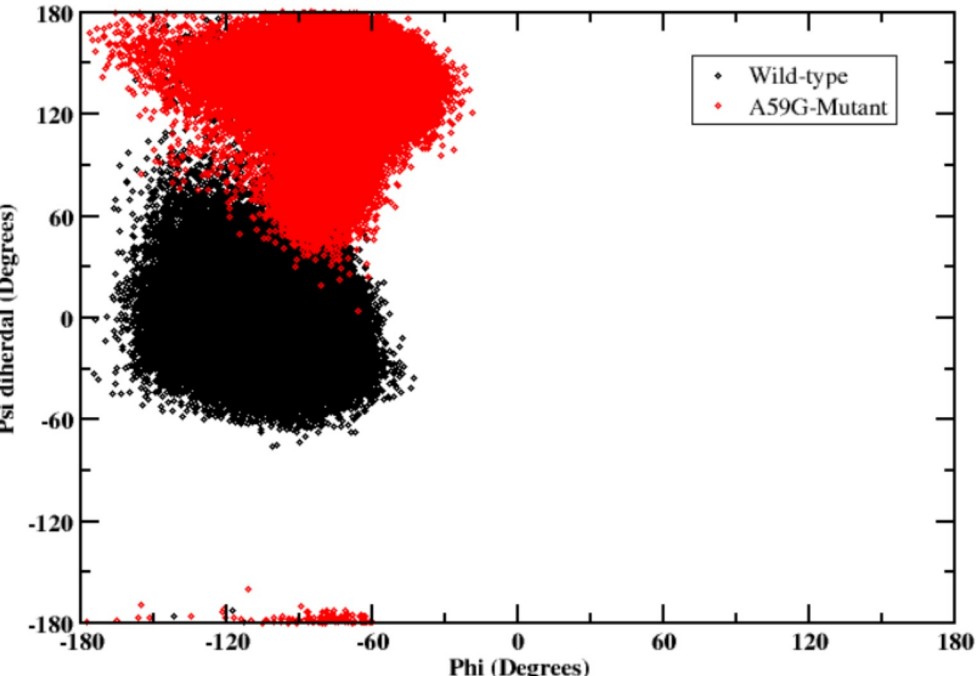

**Fig 7. 2-D plot showing the segregation of wild-type and mutant simulation trajectories, on the basis of phi(Φ) and psi(ψ) dihedrals values.** Black color depicts the wild-type and red color represents the A59G-mutant cluster.

trajectories as listed: Φ and ψ dihedrals of Gln61 and Tyr64 residues, crucial interactions between Tyr64-Tyr32, Tyr64-Thr35 and Tyr64-Gln61 and the total energy component as calculated from the MMGBSA energy analysis (S1 File). Fig 8 shows the projections of these features in the two-dimensional matrix, where the wild-type system is represented in green clusters and the mutant system in red clusters. The plot shows that the ψ dihedral of Gln61 and Tyr64 along with the distances and energy component were able to distinguish the wild-type and mutant ensembles. Fig 9 shows the projections of PC1 on PC2 for the wild-type (cyan) and A59G mutant (orange) trajectories based on the above mentioned multiple features and ~90% of the population is covered along these two PCs. The wild-type and mutant clusters showed distinct clusters especially along the PC1, as shown in Fig 9. The plot also shows that dihedrals of the residue Gln61 are negatively correlated with all the other features chosen for this multivariate PCA. Hence, these features can also prove to be an important reaction coordinate/collective variable (CV) for performing advanced biased metadynamics simulations.

### MSM based analyses of MD simulation trajectories

After performing the detailed analyses of the crucial residues of the conserved regions and the multivariate PCA, detailed insight is gained for the two systems. To further understand the origin of these distinct behaviors and to comprehensively analyze the two trajectories, Markov State Model (MSM) based analysis of the simulation trajectories was performed using PyEMMA. In principle, MSM models are the simplified coarse grained kinetic models which are constructed from the available simulation data. The MSM based analysis provided us with the distinct metastable states for the wild-type and mutant systems. The feature chosen in this MSM based kinetic models generation was "Cadist", which is the distance between

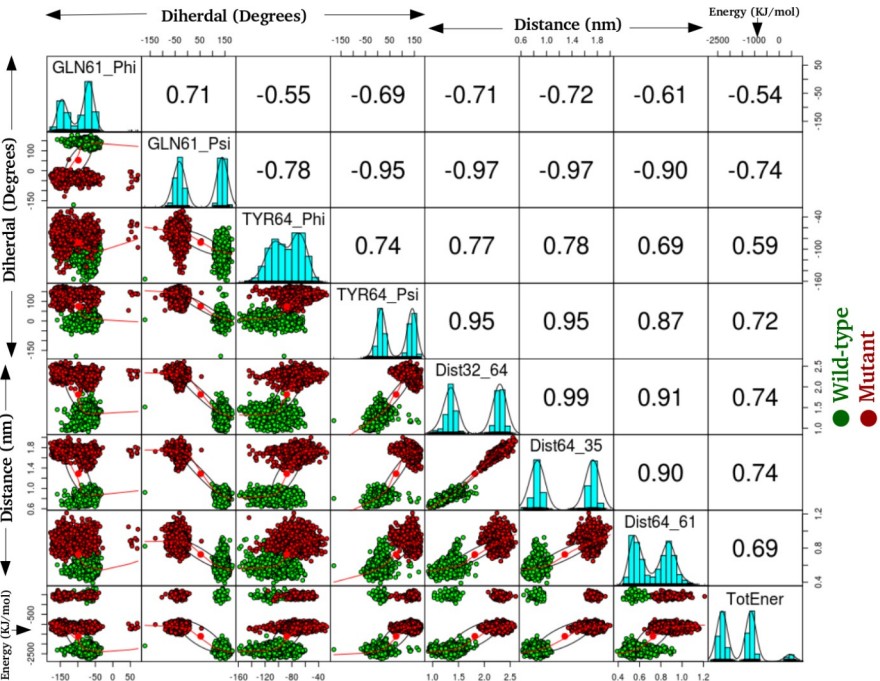

**Fig 8. The 2-D plot showing the projections of wild-type (green) and mutant (red) trajectories, using multivariate PCA approach.** A combination of dihedral, interactions and energy were used as features, as labelled on the X and Y axis.

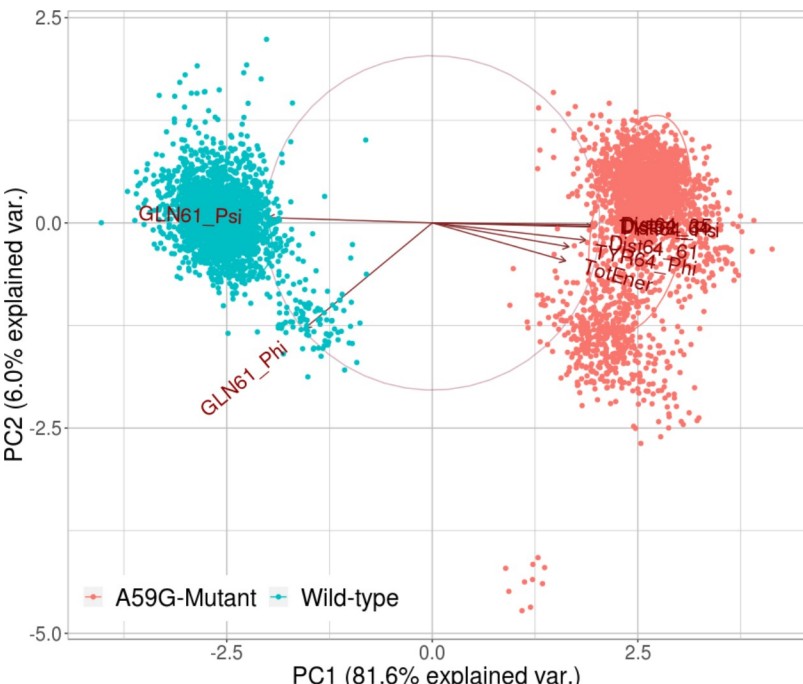

**Fig 9. Projections of PC1 on PC2, of the features chosen for multivariate PCA. Cyan and orange color clusters are for wild-type and mutant trajectories, respectively.** The individual arrows represent the eight features used for this analysis.

Cα-coordinates of the Ras protein. A lag time of 125 ns was chosen for both simulation trajectories. Fig 10(A) and 10(B) shows the schematic representation of the distinct states ensembles for the wild-type and mutant simulation trajectories using the MSM analysis. The MSM analysis of the wild-type trajectory identified four metastable states, while for the mutant trajectory five metastable states were identified, as shown in Fig 10. Fig 11(A) and 11(B) show the pathways from the rightmost to the leftmost state, for the wild-type and mutant trajectories respectively. The leftmost state is assumed to be a rare-event, while the right one is the frequently visited. The thick arrows represent the most probable and stable pathway and other splits and thin arrows demonstrate other intermediate pathways encountered for the given system. For the wild-type simulation, states 1, 2 and 3 are mostly visited, while state 4 is sparsely visited. Upon structural analysis of these states, the wild-type system was found to have similar structural characteristics. On the other hand, the mutant structure showed five metastable stables, where the metastable state 5 had altogether different characteristics than the rest of the trajectories. Metastable states 1–4 had similar structural features. Of these five states, state 5 bears most importance as it shows structural features similar to that of the wild-type counterpart. Moreover, when the transition pathways were computed for these MSM formulations, the transition time between cluster 1 (highly populated) and sparsely populated cluster 5 for the mutant system showed significant difference. It suggests that it approximately takes 3.36 millisecond to reach to the transient state in cluster 5 (wild-type like pre-hydrolysis state conformation) for the mutant, while the reverse would take 2.25 milliseconds to reach back to cluster 1. The transition values of the 5 µs of the wild ensemble are not discussed here, as other analyses show that the system did not differ in terms of active/inactive states, where most of them possess similar structural characteristics.

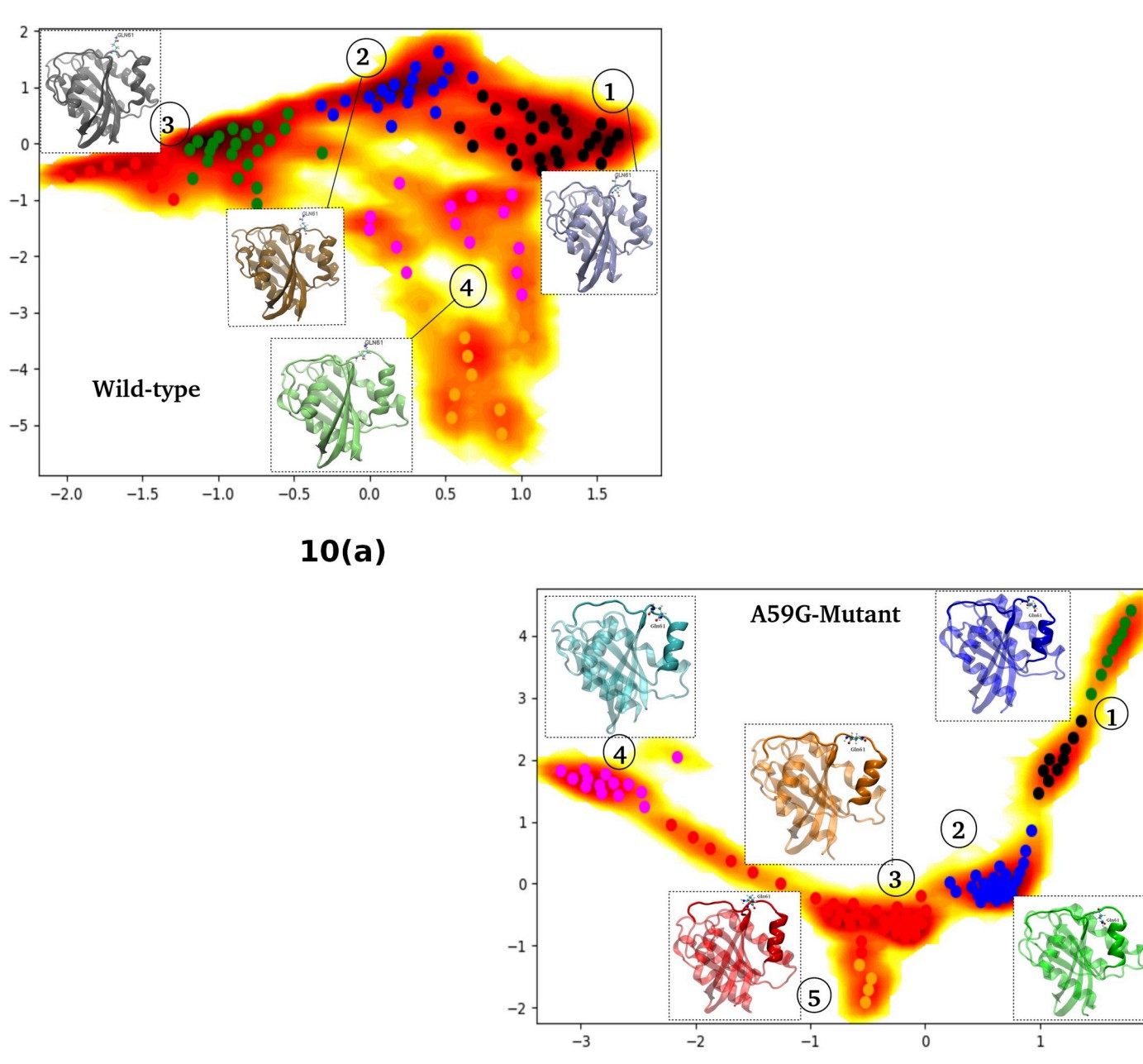

**10(a)**

**10(b)**

**Fig 10. Depiction of metastable clusters for the wild-type and mutant trajectories, represented by numbers, as captured using PyEMMA while performing MSM based analysis.** The feature used was Cadist (distances between the C-alpha atoms of the respective trajectories).

Furthermore, Fig 12 shows an overlap of the pre-hydrolysis network region of a mutant structure from the metastable state cluster 5 (yellow CPK representation) and the most stable conformations, each from wild-type simulation and mutant trajectories (cluster 1). This pre-hydrolysis state is difficult to encounter in the mutant simulations and in this 5 μs mutant simulation, the structure was observed for a short time as demonstrated by the sparsely populated cluster 5.

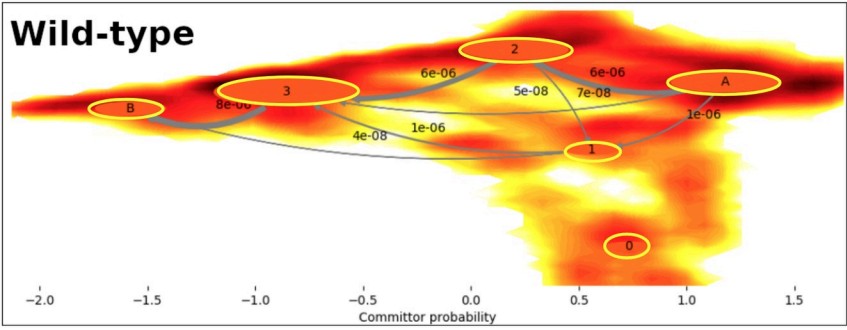

**Fig 11. The 2D-plot demonstrating the probable pathways for the wild-type and mutant simulations, calculated using PyEMMA.** The most probable pathway is denoted by a bold gray arrow (rightmost to left most cluster). The committor probability of each state transition is written along the respective arrow.

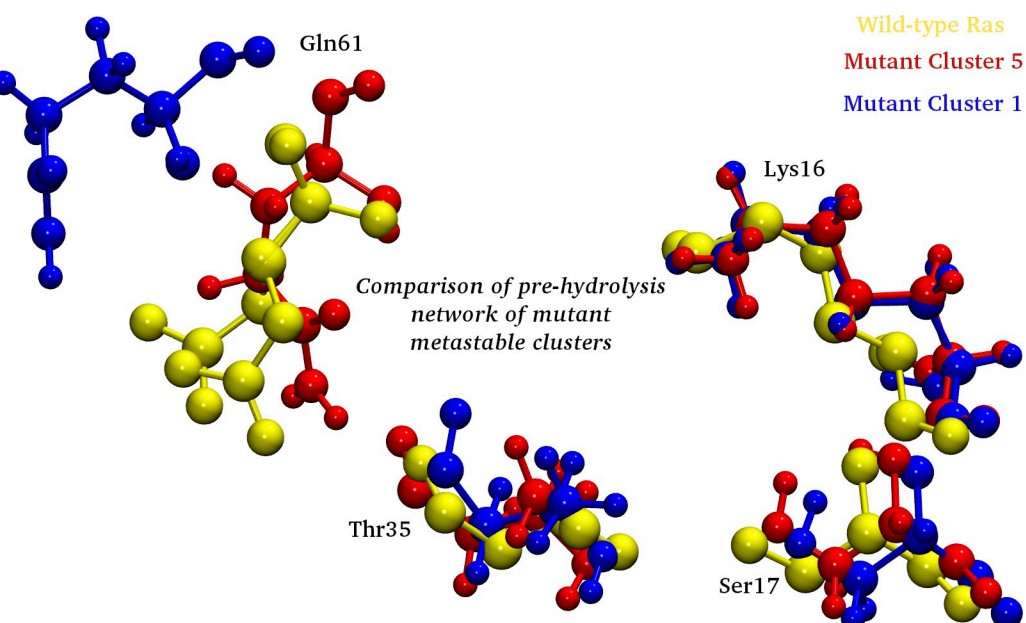

**Fig 12. Fig showing the overlap of representative structures of the cluster 5 (mutant), cluster 1 (wild-type and mutant), from the respective MSM based analysis.** Yellow CPK representation shows the mutant conformation from cluster 5 (sparsely populated) and the most probable conformations of wild-type and mutant trajectories, from the respective cluster 1.

Other additional relevant plots for the MSM analysis like VAMP score, lag-time calculation, implied time-scales and cktest are provided in the S1 File.

## Discussion

In the present study, the wild-type and mutant A59G-HRas were simulated using classical MD simulations for microseconds level time-scale, for each of the two systems. Through this study, an attempt was made to assess and characterize the distinct features of the wild-type and mutant A59G ensembles along with the comparative analyses of the pre-hydrolysis state. The absence of pre-hydrolysis state in the A59G-HRas mutant plays a pivotal role in the aberrant hydrolysis activity [48]. In this backdrop, 5 μs of classical MD simulations were carried out and in-depth analyses using advanced analytics techniques like feature-based PCA and MSM were carried out. The preliminary RMSD and RMSF bases analyses showed distinct trends of the conserved regions, especially SwII for the wild-type and mutant trajectories. The structural analyses showed remarkably different orientation of the SwII region, where the mutation (A59G) lies, which also correlates with the RMSF results. The impact of the hot-spot mutation is clearly evident on otherwise conserved regionsas well (especially SwII) [63]. The comparison also demonstrated that the regions encompassing residues 45–50 and 104–108 had considerable variance for the wild-type system, which also correlates with the simulation study on the solution structure of the H-ras p21-GTP complex [33]. The study also suggested that the loss of important intermolecular contacts upon solvation at or near the residues, might be responsible for this enhanced variation. Following this, an extended analysis was also carried out on the residues of the SwI, SwII and GBR regions. The analyses showed important residues like Tyr32, Thr35 (SwI), Gly60, Gln61 and Tyr64 (SwII) assist in maintaining their respective interactions and hence are capable while selecting features for biased simulations like umbrella sampling, steered MD, metadynamics etc. Moreover, for advanced analytics approaches like feature-based PCA (where multiple variates are required) and for MSM based techniques where CVs are required, features based on these important residues can be defined. A very recent study has also reported the importance of reaction coordinates specifically for differentiating the wild-type from G12V, T35S and Q61K mutant HRas ensembles using long 1.4μs accelerated MD simulations [64]. The study also reveals the effect of underlying mutations on the dynamic conformational changes of the conserved regions (SwI and SwII) in the mutant HRas systems. Additionally upon visual analysis, the residue Tyr64 showed quite interesting behavior in terms of side chain fluctuation and to quantify this behavior dihedral angle analysis was carried out for Tyr64 residue. A flip along the ψ dihedral of the Tyr64 residue was observed, where a complete flip from the negative (-60° to 60°) to positive quadrant (60° to 120°) was observed for wild-type and mutant ensembles, respectively. The above values for the simulation trajectories correlated well with the experimental crystal structures available at RCSB, where ψ dihedral was -24° (for wild-type pdb id: 1QRA [21] and 143° (for A59G mutant pdb id: 1LF0 [47]. This also strengthens the hypothesis about the strong impact of A59G mutation on the SwII region. Another recent study also talks about the individual yet crucial impact of specific residues on the structural arrangement for the KRas isoform [65]. The study talks about wild-type, G12C and G12D mutants of KRas and roles of specific residues like Tyr32, Gln61 and Tyr64 in destabilization of water molecules in mutant systems. Of these, Tyr32 (SwI) came out to be a central contributor in the dynamics. The quantifiable flip of the Tyr64 residue's side-chain is a crucial characteristic in A59G-HRas mutants, which also participates in making direct contacts with other important residues like Tyr32, Tyr64 and Gln61. A similar trend of SwII change and flip of Gln61 side-chain is also reported previously [25, 47, 48]. Hence, a highly pronounced effect of the A59G mutation (belonging to SwII) on the crucial

residues like Gly60, Gln61, Tyr64 is observed. The role of Gln61 residue and other residues of SwII region (Tyr64) and SwI region helps in maintaining the interaction network for GTP hydrolysis [29]. The residue Gly60's (SwII) role in GTP hydrolysis is well established [39, 41, 46]. In agreement to this, the impact of A59G mutation on neighbouring residue Gly60 is highly pronounced in the current simulation as well where 2 distinct maxima were observed for Gly60's interaction with GBR's residues Lys16 and Ser17, in mutant *simulationM*. Additionally, residues namely Tyr32, Thr35 also showed distinct wild-type and mutant behavior. In addition, the majority of the structural differences for the available Ras isoforms are localized in the C-terminal stretch and for the simulation studies in the present work we have excluded this hyper-variable C-terminal region. Furthermore, BLASTp between HRas and KRas proteins sequences shows that the residues ranging between 1–166 (i.e. excluding the hyper-variable C-terminal region) shares 94% identities and 98% similarities amongst the two isoforms (data not shown) [66]. Though the HRas isorform was used for the simulations and analyses for the present study, similar trends and behavior could also be speculated for the KRas isoform as well. A recent review on the available conformational information for the KRas isoform pointed out the role of G12D mutation in populating the metastable states for the wild-type and mutant isoforms [67]. The review outlines the experimental, structural and simulation studies on KRas in a great detail. Main emphasis on the conserved switch regions is also shed, which signifies the switch regions to be highly dynamic in KRas conformers. Further, it has also been postulated that KRas mutations affect the structure in an allosteric manner leading to changes visible at a distant site. Another study which mainly focussed on SwII region's mutants: D33E, A59G demonstrated that these conformers adopt nearly identical conformational as that of HRas [68]. These mutations induced KRas to get crystallized in open state 1 conformation. The DXXGQ motif also showed an increased flexibility across the SwII rearrangement for the A59G KRas mutants. After understanding and enlisting the crucial distinguishing features of the two systems, a multivariate PCA analysis was performed using the R statistical package. The main aim of performing this feature-based PCA is dimensionality reduction, which could give more refined features for the biased simulations like steered MD, metadynamics as well for feature-based analytics package like MSM based analysis tool for example PyEMMA. In the present study, a total of eight features were chosen for this namely: Gln61 and Tyr64 $\Phi$ and $\Psi$ dihedrals, distances between Tyr32-Tyr64, Tyr64-Thr35, Tyr64-Gln61 and total energy calculated using MMGBSA. The selected features could segregate the wild-type and mutant trajectories and hence can be used as reaction coordinate/CV for featured based simulations or analyses. The study also observed that the dihedrals of the residue Gln61 are negatively correlated with the rest of the features. To further understand the origin of these distinct behaviors and to comprehensively analyze the two trajectories, Markov State Model (MSM) based analysis of the simulation trajectories was performed using PyEMMA. MSM based analyses can help in systematically re-define the simulation data from long MD simulations in few important metastable structures [56]. Four metastable were observed for the wild-type simulation and five for A59G mutant simulation. The MSM based analysis that resulted in four broad metastable clusters for wild-type, showed little differences in the structures of these states. For the mutant simulation, cluster 5 was an interesting finding, where few structures with the interaction network characteristic of pre-hydrolysis state were also observed, as depicted in the overlap of structures from this cluster with respect to one of the wild-type states.

The present study reconciles that the SwII region is highly influenced by the neighboring A59G mutation and a great degree of restructuring occurs due to this mutation. The absence of pre-hydrolysis state in the mutant conformation is the major reason for this, where the crucial bonding network is not present. The subsequent MSM analyses resulted in few

energetically expensive pre-hydrolysis states even in the mutant simulations. This suggests that the absence of pre-hydrolysis state in the mutant system, required for regular cycling of active-inactive state transition in Ras isoforms, reasons for its permanently active oncogenic state.

## Conclusion

The present study focussed on exploring the wild-type and mutant A59G-HRas systems with a perspective of enlisting crucial distinguishing features that could classify and segregate the two systems, in classical or biased MD simulations. The overlap of the available crystal structures of the wild-type and mutant Ras showed visible difference in SwII orientation, especially in side-chain orientation of Gln61 and Tyr64 residues. In the wild-type crystal structure (1QRA), the side chains of these residues point inwards, towards GTP and Mg (i.e. towards the pre-hydrolysis network region). On the other hand, mutant conformation had exactly the opposite orientation of these residues pointing towards the solvent. The MD simulations of these con-formations also showed the same trend, which was maintained for the 5 μs of the respective simulation. The preliminary analyses on these trajectories demonstrated clear discrimination between the SwII region, along with few residue stretches between 25–29, 45–50 and 104–108. Structural analyses of the two trajectories were also carried out to quantify the changes in these residue stretches, as depicted by the relative thickness in the three-dimensional structural over-laps of the two systems. The orientation of SwII and thickness of SwII along with residues 25–29 and 45–50 was clearly observed in this overlap. The findings of basic RMSD, RMSF analyses complement with the crucial interactions of the important residues from conserved SwI, SwII and GBR regions. Residues Gly60, Gln61, Tyr64 (all belonging to SwII region) showed distinct interaction trends with important residues of SwI and GBR. The dihedral angle analysis of Tyr64 was also calculated as this residue showed major differences in its respective orientations in the crystal structure overlaps. MMGBSA calculation was also done for the two trajectories and the coulombic and polar solvation energy components showed a difference of ~1000 KJ/mol and ~1400 KJ/mol between the wild-type and mutant systems. Combining the outcomes of the above discussed preliminary analyses, a multivariate PCA was carried out using a combi-nation of eight features. The features like ψ dihedral of Gln61 and Tyr64 along with interac-tions of Tyr64 residue (with Tyr32, Thr35 and Gln61) and total energy component from MM-GBSA analysis were chosen for this PCA. Interestingly, Φ and ψ dihedrals of Gln61 nega-tively correlated with the other six variates. To summarize, on the basis of these eight features, the mutant and wild-type systems could be clearly differentiated. This suggests that an optimal combination of any of these eight features could be used as reaction coordinate/CV for meta-dynamics, steered MD, umbrella sampling like biased simulation. Moreover, advanced analy-ses methods like MSM based techniques of kinetic model generation and thermodynamics assessment could also be used taking into account the above listed features. As feature selection is a key factor in advanced MD simulations, these features might prove useful while selecting the CVs. Lastly, MSM based analysis of the individual trajectories was also carried out using Cadist as the selected feature. The main finding of this analysis was observed in the mutant simulation, where a sparsely populated cluster of energetically expensive metastable states was observed where pre-hydrolysis interacting network was observed. This pre-hydrolysis network is essentially a feature of wild-type ensemble, where the regular switching of active and inactive state conformation is intact and hence is not observed in mutant ensembles as they tend to be stuck in permanently active state. This wild-type-like conformation (having pre-hydrolysis conformation) was missed while performing classical analyses for the 5 μs mutant simulation trajectory, being very few in number, but MSM analyses could capture this state as well. Upon reconciliation of the basic and advanced analyses and correlating the same with existing

experimental evidence available, the impact of A59G mutation of nearby residues like Gln61 and Tyr64 is established, which also plays a crucial role in GTP hydrolysis.

To conclude, the present study postulates the role of A59G mutation in destabilizing the pre-hydrolysis state normally observed in the wild-type RAS state. MSM analytics could capture the very short lived pre-hydrolysis state in case of A59G mutation. Hence, drug/inhibitor docking and re-purposing studies with this Ras mutant might help to bring back stability to the pre-hydrolysis state essential for the hydrolysis of Ras.

## Supporting information

**S1 File.**
(DOCX)

## Acknowledgments

This work was performed using the "Bioinformatics Resources and Applications Facility (BRAF)" and "National PARAM Supercomputing Facility (NPSF)" at C-DAC, Pune.

## Author Contributions

**Conceptualization:** Neeru Sharma, Uddhavesh Sonavane.

**Data curation:** Neeru Sharma.

**Formal analysis:** Neeru Sharma.

**Investigation:** Neeru Sharma, Uddhavesh Sonavane.

**Methodology:** Neeru Sharma, Uddhavesh Sonavane.

**Supervision:** Rajendra Joshi.

**Validation:** Uddhavesh Sonavane.

**Visualization:** Neeru Sharma.

**Writing – review & editing:** Neeru Sharma, Uddhavesh Sonavane, Rajendra Joshi.

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
