## [Decision Letter · Decision Letter 0]

9 Aug 2020

PONE-D-20-16543

Comparative MD simulations and advanced analytics based studies on wild-type and hot-spot mutant A59G HRas

PLOS ONE

Dear Dr. Joshi,

Thank you for submitting your manuscript to PLOS ONE. After careful consideration, we feel that it has merit but does not fully meet PLOS ONE’s publication criteria as it currently stands. Therefore, we invite you to submit a revised version of the manuscript that addresses the points raised during the review process.

The ideas are interesting, but there are several shortcomings and inconsistencies within the paper and with your previous work that should be addressed before reassessing suitability for publication. 

We look forward to receiving your revised manuscript.

Kind regards,

Paul A Randazzo

Academic Editor

PLOS ONE

Journal Requirements:

Reviewers' comments:

Reviewer's Responses to Questions

**Comments to the Author**

1. Is the manuscript technically sound, and do the data support the conclusions?

Reviewer #1: Yes

Reviewer #2: Partly

2. Has the statistical analysis been performed appropriately and rigorously? 

Reviewer #1: N/A

Reviewer #2: N/A

3. Have the authors made all data underlying the findings in their manuscript fully available?

Reviewer #1: Yes

Reviewer #2: Yes

4. Is the manuscript presented in an intelligible fashion and written in standard English?

Reviewer #1: Yes

Reviewer #2: No

5. Review Comments to the Author

Reviewer #1: Using molecular dynamics simulation, and cutting edges computational analysis such as MSM based analytics, multivariate PCA analysis, and MM-GBSA energy calculation, this paper tests the hypothesis that mutant A59G as a single residue substitution in H-Ras protein can have significant effects on the protein conformation, dynamics, and pre-GTP hydrolysis network. The authors conducted a cumulative ~10 μs MD simulations comparing the A59G-Ras mutant protein to its wild-type. They examined the conformational changes for the wild-type and mutant by Markov State Model (MSM). The authors delineated the roles of critical residues like Tyr32, Thr35, Gln61, and Tyr64 using multivariate PCA analyses. Besides, they also computed the interaction between relevant vital residues and the highly conserved regions such as P-loop, Switch I, and Switch II.

The manuscript is well written, clear, precise, and easy to understand. In general, I find this study to make a valuable contribution to Ras scientific literature. I recommend the manuscript for publication after some adjustments have been taken into accounts.

Specific Remarks:

1. It is puzzling why authors did not choose K-Ras for MD simulation, could authors speculate what happens to K-Ras mutant A59G by this MD simulation in the discussion section?

2. How many independent simulations had been conducted for Figure 1?

3. In Figures 4, 5, and 6, the labels in the charts are vague. Need to label each of these figures with a, b, c, etc. for the clarity. The same for Figures 10 and 11, authors need to label Figure 10a and Figure 10b, Figure 11a, and Figure 11b. It’s hard to see the number labeling in Figure 11.

4. All the figure legends (Figure 1-12) are scattering among the results section. Figure legends should be consolidated together.

5. The statements on line 235-237 “The GBR and SwI region did not show major deviation, with respective RMSD values of 3 Å and 4 Å for GBR and 0.75 Å and 0.5 Å for SwI (for wild-type and mutant systems, respectively).” are not consistent with the data?

Reviewer #2: Mutations of RAS proteins play a causal role in human cancer and computational approaches are an important tool to study the role of RAS conformational dynamics in the regulation of GTP hydrolysis. This manuscript by Sharma N, et al. investigated the role of the A59G mutation to form A59GHRas on the conformational dynamics of HRas using µs long all atoms MD simulations. This work is a follow up to the previous work by Sharma, N, et al (Comput. Biol. Chem., 2017), where the dynamics of the Switch II region (SwII region) is suggested to play a role in the formation of the GTP pre hydrolysis state. Here, the authors show that the conformation and the dynamics of the SwII region differ in the WTHRas and A59GHRas, with a detailed study of the role of Tyr64. The authors also performed a Markov State Model analysis of the trajectories calculated for WTHRas and A59GHRas which suggests that the GTP pre hydrolysis conformational state is sparsely populated in the case of A59GHRas. As a PLOS one, I don’t rank this paper in the highest category and I think the paper needs to undergo significant improvements before getting published. Below is a list of some of the points that lack clarity.

1- In this paper, the authors show difference between the RMSD and RMSF of WTHRas and A59GHRas, in particular in the very relevant SwII region. Fig. 2. shows a reduction of the fluctuations for residues 45-50 and 104-108, and a 5-residue shift in the magnitude of fluctuations for residues 55-68, which are part of SwII. This is interesting. However, this appears to be somewhat different from has been observed in the Sharma, N, et al paper (Comput. Biol. Chem., 2017, Fig5 and S9a, b) for the same type of calculations. I think the authors should clarify the differences, or if the analysis has been different in the two studies.

2- In order to characterize the interactions between functionally important regions of the protein, the authors compared distances between a set of residues in Fig 4, 5 and 6. Fig. 5 shows the distance between G60 and K16/K17 binned as a function of the number of conformers. I think the authors could comment on the fact that two maxima are observed. Is this relevant in the context of GTP hydrolysis?

3- The functional role of conformational dynamics is well established, if not perfectly understood, yet I think the discussion could be very stimulating if results presented here for A59GHRas were to be compared to results that are available in the literature for KRas and mutants, in terms of conformational flexibility.

6. PLOS authors have the option to publish the peer review history of their article (what does this mean?). If published, this will include your full peer review and any attached files.

Reviewer #1: No

Reviewer #2: No

---

## [Author Response · Author response to Decision Letter 0]

24 Sep 2020

Response to Reviewers

Reviewer #1: 

Comment 1: It is puzzling why authors did not choose K-Ras for MD simulation, could authors speculate what happens to K-Ras mutant A59G by this MD simulation in the discussion section?

Response: The H-Ras isoform was preferred for this work than its KRas counterpart because at the time of starting this work, only the crystal structure of wild-type HRas isoform was available (excluding the hyper-variable C-terminal region). Also, the majority of the structural differences for the available Ras isoforms are localized in the C-terminal stretch and for the simulation studies in the present work we have excluded this hyper-variable C-terminal region. Furthermore, BLASTp shows that the residues ranging between 1-166 (i.e. excluding the hyper-variable C-terminal region) shares 94% identities and 98% similarities amongst the two isoforms [66] [Altschul SF, Gish W, Miller W, Myers EW, Lipman DJ. Basic local alignment search tool. J Mol Biol. 1990 Oct 5;215(3):403-10. doi: 10.1016/S0022-2836(05)80360-2. PMID: 2231712]. So, it can also be speculated that the KRas systems (wild-type and A59G mutant) would also depict similar trends as that of HRas simulations. The same has also been added in the discussion section along with the references from studies where KRas was used [Page 19, line numbers 471-488 of the revised manuscript].

Comment 2: How many independent simulations had been conducted for Figure 1?

Response: For Fig 1, two independent MD simulations were performed, one each for wild-type and mutant A59G HRas systems. Each simulation, namely simulationW (wild-type HRas) and SimulationM (A59G HRas) were carried out for 5µs each. The same is now explicitly mentioned in the revised manuscript to avoid any further confusion [Page 7, line numbers 161-163 of the revised manuscript].

Comment 3: In Figures 4, 5, and 6, the labels in the charts are vague. Need to label each of these figures with a, b, c, etc. for the clarity. The same for Figures 10 and 11, authors need to label Figure 10a and Figure 10b, Figure 11a, and Figure 11b. It’s hard to see the number labeling in Figure 11.

Response: All the figures have been modified and updated as per the reviewer’s comments and labelled separately as directed. Figure 10 (a) and 10 (b) have also been updated with the representative structural snapshots from distinct MSM metastable clusters to provide more information. Similarly for figure 11 (a) and 11 (b), individual states (0, 1, 2, 3, A and B) have been highlighted in yellow ovals for better visualization along the pathways.

Comment 4: All the figure legends (Figure 1-12) are scattering among the results section. Figure legends should be consolidated together.

Response: The journal submission guidelines required the figure legends to be included in the main manuscript where the figure is supposed to be included in the print version and not as a separate document. The guideline states “Place figure captions in the manuscript text in read order, immediately following the paragraph where the figure is first cited. Do not include captions as part of the figure files or submit them in a separate document”. That is the reason, we had to include the figure legends in the manuscript which would be placed along with the figures at the time of final publishing by the journal. 

Comment 5: The statements on line 235-237 “The GBR and SwI region did not show major deviation, with respective RMSD values of 3 Å and 4 Å for GBR and 0.75 Å and 0.5 Å for SwI (for wild-type and mutant systems, respectively).” are not consistent with the data?

Response: The authors have corrected the text for the mentioned inconsistency. The RMSD values for GBR were mis-typed (the correct values being 0.25 Å and 0.5 Å respectively for wild-type and mutant systems) and have now been corrected in the revised manuscript [Page 10, line numbers 238-240 of the revised manuscript].

Reviewer #2: 

Comment 1: In this paper, the authors show difference between the RMSD and RMSF of WTHRas and A59GHRas, in particular in the very relevant SwII region. Fig. 2. shows a reduction of the fluctuations for residues 45-50 and 104-108, and a 5-residue shift in the magnitude of fluctuations for residues 55-68, which are part of SwII. This is interesting. However, this appears to be somewhat different from has been observed in the Sharma, N, et al paper (Comput. Biol. Chem., 2017, Fig5 and S9a, b) for the same type of calculations. I think the authors should clarify the differences, or if the analysis has been different in the two studies.

Response: The simulations performed for this study are different from the ones reported in [48] [Sharma N, Sonavane U, Joshi R. Differentiating the pre-hydrolysis states of wild-type and A59G mutant HRas: An insight through MD simulations. Comput Biol Chem. 2017 Aug;69:96-109. doi: 10.1016/j.compbiolchem.2017.05.008. Epub 2017 Jun 1. PMID: 28600956]. The authors have tried to replicate the same simulation conditions, factors and environment in order to see the consistency of the crucial features distinguishing wild-type and A59G mutant systems. The only difference being that the start structures of these simulations were captured from the metadynamics simulations (from the wild-type and mutant clusters) as reported in the previous study. The same has now been clearly stated in the ‘Materials and Methods’ section of the manuscript to avoid any further confusion for the reviewers and readers [Page 7, line numbers 173-175 of the revised manuscript]. While most of the features including the ones from the conserved GEF-Binding Region (GBR), SwI and SwII were in agreement with the previous set of results, few of the features (SwII RMSF and flexibility) showed a different trend too, as highlighted in Fig 2 of the manuscript. In the previous paper by the same authors, the main focus was on emphasizing the role of water in demonstrating the importance of pre-hydrolysis state in wild-type Ras and its absence in the mutant counterpart. To demonstrate the same, interactions of residues from the GEF-Binding Region (Lys16, Ser17), SwI (Thr35) and SwII (Asp57, Gly60, Gln61) were reported with γ phosphate and Mg. We have calculated all these interactions for the present simulation trajectories as well and they were in agreement with the previous results (not shown in the manuscript). In this manuscript, the authors have tried to report additional distinguishing features of the wild-type and mutant systems by performing both traditional and advanced analyses which includes distances between residues of the conserved regions, MSM based analyses taking crucial features as the reaction coordinates, PCA etc. Further, capturing the structural features of SwI and SwII is already a known challenge owing to the high flexibility of switch regions, even for crystallography / NMR experiments. Hence, the role of MD simulations in capturing various ensembles states of KRas is very crucial [67] [Pantsar T. The current understanding of KRAS protein structure and dynamics. Comput Struct Biotechnol J. 2019 Dec 26;18:189-198. doi: 10.1016/j.csbj.2019.12.004. PMID: 31988705; PMCID: PMC6965201.]. The present study has helped us to understand the impact of A59G mutation in the GTP hydrolysis due to distinct trends of the crucial residues like Gln61 and Tyr64 (of the conserved SwII region) in the wild-type and mutant simulations.

Comment 2: In order to characterize the interactions between functionally important regions of the protein, the authors compared distances between a set of residues in Fig 4, 5 and 6. Fig. 5 shows the distance between G60 and K16/K17 binned as a function of the number of conformers. I think the authors could comment on the fact that two maxima are observed. Is this relevant in the context of GTP hydrolysis?

Response: The interaction between residues Gly60 (SwII) with Lys16 and Ser17 (GBR) as depicted in Fig 5 was shown to specifically emphasize the differences in the respective distances between residues belonging to SwII and GBR. The direct role of Gly60 in GTP hydrolysis is well established as shown in the references [39], [41], [46] of the revised manuscript. So, in addition to its known interaction with γ phosphate and water, we tried to demonstrate the same with its Lys16 and Ser17 interactions and found interesting trends for Gly60-Lys16 and Gly60-Ser17 interactions. The fact that two maxima are observed is also owing to the fact that Gly60 lies adjacent to the mutant residue (A59G) and glycine itself being the simplest amino acid, it showed a lot of flexibility and variation in the mutant simulations and hence the trend in Fig 5. Furthermore, this has more to do with Gly60 position, its respective location in the mutant A59G structure, than with Lys16 and Ser17. Gly60 showed an increased fluctuation for the mutant simulation as compared to the wild-type HRas simulation, which can also be seen in the RMSF plot in Fig 2 and Fig R1 of Response to Reviewers document. Both Lys16 and Ser17 maintained a stable interaction with Mg but Gly60-GTP interaction showed fluctuation in the mutant simulations. To clarify the same in the manuscript, we have added an explanation in the results section where this figure is discussed [Page 12, line numbers 303-305 and Page 13, line numbers 309-313 of the revised manuscript] and discussion section [Page 19, line numbers 467-470 of the revised manuscript]. To depict this pictorially for the reviewer, we have plotted a scatter plot between Gly60-γ phosphate distance (X-axis) and ψ dihedral of Gly60 (Y-axis), which shows the wide range of Gly60-γ phosphate distance for the A59G mutant simulation. Please find the fig R1 for your ready reference, in response to reviewers document.

Comment 3: The functional role of conformational dynamics is well established, if not perfectly understood, yet I think the discussion could be very stimulating if results presented here for A59G-HRas were to be compared to results that are available in the literature for KRas and mutants, in terms of conformational flexibility.

Response: As per the reviewer’s suggestion, we have modified the discussion section to give a better insight in terms of conformational flexibility by comparing the results with the available and the latest literature on Ras [Page 18, line numbers 445-449, Page 18, line numbers 456-460, Page 19, line numbers 471-488 of the revised manuscript].

---

## [Decision Letter · Decision Letter 1]

6 Oct 2020

Comparative MD simulations and advanced analytics based studies on wild-type and hot-spot mutant A59G HRas

PONE-D-20-16543R1

Dear Dr. Joshi,

We’re pleased to inform you that your manuscript has been judged scientifically suitable for publication and will be formally accepted for publication once it meets all outstanding technical requirements.

Kind regards,

Paul A Randazzo

Academic Editor

PLOS ONE

Additional Editor Comments (optional):

Reviewers' comments:

Reviewer's Responses to Questions

**Comments to the Author**

1. If the authors have adequately addressed your comments raised in a previous round of review and you feel that this manuscript is now acceptable for publication, you may indicate that here to bypass the “Comments to the Author” section, enter your conflict of interest statement in the “Confidential to Editor” section, and submit your "Accept" recommendation.

Reviewer #1: All comments have been addressed

Reviewer #2: All comments have been addressed

2. Is the manuscript technically sound, and do the data support the conclusions?

Reviewer #1: Yes

Reviewer #2: Yes

3. Has the statistical analysis been performed appropriately and rigorously? 

Reviewer #1: N/A

Reviewer #2: Yes

4. Have the authors made all data underlying the findings in their manuscript fully available?

Reviewer #1: Yes

Reviewer #2: Yes

5. Is the manuscript presented in an intelligible fashion and written in standard English?

Reviewer #1: Yes

Reviewer #2: Yes

6. Review Comments to the Author

Reviewer #1: I have no further comments at this point that would improve its quality and I therefore recommend publication.

Reviewer #2: (No Response)

7. PLOS authors have the option to publish the peer review history of their article (what does this mean?). If published, this will include your full peer review and any attached files.

Reviewer #1: No

Reviewer #2: No

---

## [Editor Report · Acceptance letter]

8 Oct 2020

PONE-D-20-16543R1 

Comparative MD simulations and advanced analytics based studies on wild-type and hot-spot mutant A59G HRas 

Dear Dr. Joshi:

I'm pleased to inform you that your manuscript has been deemed suitable for publication in PLOS ONE. Congratulations! Your manuscript is now with our production department. 

Kind regards, 

on behalf of

Dr. Paul A Randazzo 

Academic Editor

PLOS ONE